Comparative analyses of olfactory systems in terrestrial crabs (Brachyura): evidence for aerial olfaction?

Krieger Jakob 1
Braun Philipp 1
Rivera Nicole T. 2
Schubart Christoph D. 2
Müller Carsten H.G. 3
Harzsch Steffen 1 steffen.harzsch@uni-greifswald.de
1 Zoological Institute and Museum, Department of Cytology and Evolutionary Biology, Ernst-Moritz-Arndt Universität Greifswald , Greifswald , Germany
2 Institute for Zoology, Department of Zoology & Evolution, Universität Regensburg , Regensburg , Germany
3 Zoological Institute and Museum, Department of General and Systematic Zoology, Ernst-Moritz-Arndt Universität Greifswald , Greifswald , Germany
Williams Darren
Electronic publication date: 2015 Dec 22
Publication date: 2015
Volume: 3
Electronic Location ID: e1433
Received 2015 Sep 12; Accepted 2015 Nov 3
Copyright: ©2015 Krieger et al.
Copyright year: 2015
Copyright holder: Krieger et al.
License: This is an open access article distributed under the terms of the Creative Commons Attribution License, which permits unrestricted use, distribution, reproduction and adaptation in any medium and for any purpose provided that it is properly attributed. For attribution, the original author(s), title, publication source (PeerJ) and either DOI or URL of the article must be cited.
License URL: https://creativecommons.org/licenses/by/4.0/

Keywords: Land crabs, Terrestrialization, Decapods, Neuroanatomy, Immunohistochemistry, Morphometry, Sexual dimorphism, Crustaceans

Funding: The authors received no funding for this work.

==============================
Adaptations to a terrestrial lifestyle occurred convergently multiple times during the evolution of the arthropods. This holds also true for the “true crabs” (Brachyura), a taxon that includes several lineages that invaded land independently. During an evolutionary transition from sea to land, animals have to develop a variety of physiological and anatomical adaptations to a terrestrial life style related to respiration, reproduction, development, circulation, ion and water balance. In addition, sensory systems that function in air instead of in water are essential for an animal’s life on land. Besides vision and mechanosensory systems, on land, the chemical senses have to be modified substantially in comparison to their function in water. Among arthropods, insects are the most successful ones to evolve aerial olfaction. Various aspects of terrestrial adaptation have also been analyzed in those crustacean lineages that evolved terrestrial representatives including the taxa Anomala, Brachyura, Amphipoda, and Isopoda. We are interested in how the chemical senses of terrestrial crustaceans are modified to function in air. Therefore, in this study, we analyzed the brains and more specifically the structure of the olfactory system of representatives of brachyuran crabs that display different degrees of terrestriality, from exclusively marine to mainly terrestrial. The methods we used included immunohistochemistry, detection of autofluorescence- and confocal microscopy, as well as three-dimensional reconstruction and morphometry. Our comparative approach shows that both the peripheral and central olfactory pathways are reduced in terrestrial members in comparison to their marine relatives, suggesting a limited function of their olfactory system on land. We conclude that for arthropod lineages that invaded land, evolving aerial olfaction is no trivial task.

Introduction

Land-living crustaceans are fascinating animals that adapted during a relatively short evolutionary time period to a number of highly diverse terrestrial habitats in which they have become highly successful, and in some cases the predominant life forms (Hansson et al., 2011). Representatives in not less than five major malacostracan crustacean taxa have conquered the terrestrial habitat independently. Because the successful transition from a marine or freshwater habitat to terrestrial life requires a number of physiological adaptations which are important for survival out of water, terrestrial crustaceans constitute an excellent animal group to study evolutionary adaptations related to the invasion of land. Such adaptations include changes to gas exchange, salt and water balance, nitrogenous excretion, thermoregulation, molting, and reproduction (reviews in Bliss & Mantel, 1968; Burggren & McMahon, 1988; Greenaway, 1988; Greenaway, 1999; Greenaway, 2003; McMahon & Burggren, 1988; Powers & Bliss, 1983). The Brachyura (short-tailed crabs or “true crabs”) include several lineages that invaded land. The degree of terrestrial adaptation in crustaceans has been categorized into five classes ranging from T1 to T5 depending on the degree of independence from immersion in water and the animal’s need to access water for reproduction (Hartnoll, 1988; Greenaway, 1999; Powers & Bliss, 1983; but see Schubart et al., 2000 for an alternative classification). In this traditional classification, several brachyuran taxa have been ranked within the two highest grades of terrestrial adaptation (e.g., Gecacinidae, and some representatives of the Sesarmidae, Potamidae, Gecarcinidae, Potamonautidae, Pseudothelphusidae, and Trichodactylidae), whereas many amphibious freshwater forms and supra-littoral species are ranked in less terrestrialized categories.

The phylogenetic relationships of Brachyura and the systematics of brachyuran taxa are the topic of ongoing research (Scholtz & Richter, 1995; Dixon, Ahyong & Schram, 2003; Ahyong & O’Meally, 2004; Ng, Guinot & Davie, 2008; Tsang et al., 2014; Brösing, Richter & Scholtz, 2007; Fig. 1) but there is increasing evidence that the conquest of land occurred several times independently amongst Brachyura as suggested by Powers & Bliss (1983) and Hartnoll (1988). All members of the Gecarcinidae (with the exception of the genus Epigrapsus) and representatives of the grapisd genus Geograpsus have achieved complete terrestriality as adults, but larval development, which is not abbreviated, takes place in the oceans. The family Sesarmidae (sensu Schubart, Cuesta & Felder, 2002) includes crabs such as Sesarma jarvisi, S. cookei, and S. verleyi, which radiated into a broad range of terrestrial habitats, including mountainous rain forest and caves on Jamaica (Wolcott, 1988; Schubart, Diesel & Hedges, 1998; Diesel & Schubart, 2000; Diesel, Schubart & Schuh, 2000). The bromeliad crab Metopaulias depressus raises its offspring in water-filled leaf axils of bromeliads and certainly has evolved one of the most notable reproductive adaptations to terrestrial habitats (Diesel & Schubart, 2000), but remains immersed in water for extended time periods. The Ocypodidae comprise the genera Ocypode, Uca, and Ucides; and some of its representatives were qualified to reach terrestrial grade T3 by Powers & Bliss (1983). In the phylogenetic analysis based on stomach ossicles by Brösing, Richter & Scholtz (2007), several taxa with terrestrial tendencies, the Potamonautidae, Ocypodidae, Gecarcinidae, Grapsidae, and Mictyridae cluster together with other taxa in the proposed taxon Neobrachyura, suggesting a close relationship of those brachyuran groups which include terrestrial forms, but this grouping is not recovered in the newest and most comprehensive phylogeny by Tsang et al. (2014), so that it appears to be based on convergences. The paraphyletic superfamily Grapsoidea (comprising 88 genera with over 480 species including the Gecarcinidae (6 genera with 19 species) include intertidal to supratidal as well as limnic forms in addition to terrestrial ones, so that there is increasing evidence that the colonization of inland habitats evolved in several lineages (Schubart et al., 2000; Schubart et al., 2006; Tsang et al., 2014).

Figure 1 Simplified phylogenetic relationships among Brachyura after Tsang et al. (2014).

For simplification some major brachyuran clades are grouped. Note that brachyuran clades including representatives showing higher degrees of terrestrial adaptation (Ti > T2) are indicated by orange circles. Groups of investigated species are highlighted by indented and larger (bold) letters and are color-coded according to their lifestyles (pale blue, freshwater crabs; dark blue, marine; brown, terrestrial).

An essential physiological adaptation to master a terrestrial lifestyle during and after an evolutionary transition from sea to land includes the need for sensory organs to function in air instead of in water (Greenaway, 1999; Greenaway, 2003; Hansson et al., 2011). Mechanosensory systems must detect stimuli that propagate in air versus in water, and visual systems must operate in media with different refractive properties. In olfaction, a transition from sea to land means that molecules need to be detected in or bound from gas phase instead of being transmitted directly from one water solution (e.g., sea water) into another one (receptor lymphs). Marine crustaceans live in a world full of chemical information. It is well established that they use chemical cues to locate mates, signal dominance, recognize individual conspecifics, find favored food and appropriate habitats, and assess threats such as the presence of predators (reviews e.g., Derby et al., 2001; Grasso & Basil, 2002; Derby & Sorensen, 2008; Hazlett, 2011; Thiel & Breithaupt, 2011; Wyatt, 2011; Derby & Weissburg, 2014). However, aquatic versus land-living animals must detect highly different semiochemicals, because the medium places different demands on the compounds used. In water, molecules have to be more or less water-soluble and stable enough to travel from one individual to another. On land, semiochemicals have to be light enough to form a gas in the ambient temperatures where animals live (discussed in Stensmyr et al., 2005). These molecules also have to be sufficiently chemically stable to reach the sensory receptor cells. These new selection pressures take part together in reshaping the sense of smell during the invasion of new, terrestrial habitats (reviews Hansson et al., 2011; Hay, 2011; Weissburg, 2011).

Malacostracan crustaceans living in aquatic habitats use several systems for detecting chemicals, and these are distributed over their entire body surface, walking appendages, and mouthparts, but are also concentrated on two pairs of antennae (reviews e.g., Hallberg, Johansson & Elofsson, 1992; Hallberg & Skog, 2011; Schmidt & Mellon, 2011). The first antennal pair (the antennules) is equipped with specialized olfactory sensillae (aesthetascs) in addition to bimodal chemo- and mechanosensilla (contact chemoreceptors), whereas the second pair of antennae is only equipped with the latter. The tips of the first antennae (more specifically the lateral flagella) bear a tuft region with arrays of aesthetascs that house branched dendrites of olfactory sensory neurons (reviews by Hallberg, Johansson & Elofsson, 1992; Hallberg & Hansson, 1999; Mellon Jr, 2007; Hallberg & Skog, 2011; Schmidt & Mellon, 2011; Derby & Weissburg, 2014). Schmidt & Mellon (2011) pointed out that in aquatic crustaceans, chemical information is received and processed in two fundamentally different modes. The first mode is “olfaction” defined as chemoreception mediated by the aesthetasc pathway; the second mode is called “distributed chemoreception” defined as chemoreception mediated by contact chemoreceptors on all appendages (Schmidt & Mellon, 2011). Chemosensory neurons associated with the aesthetascs versus the contact chemoreceptors on the first antenna of malacostracan crustaceans innervate distinct regions in the brain. The axons of the olfactory sensory neurons associated with the aesthetascs target the deutocerebral chemosensory lobes (DCLs; also called olfactory lobes), whereas the axons associated with non-aesthetasc sensilla innervate the lateral antenna 1 neuropil (LAN; Schachtner, Schmidt & Homberg, 2005; Schmidt & Mellon, 2011; Strausfeld, 2012; Loesel et al., 2013). As for the different functions of these two modes of aquatic chemoreception, Schmidt & Mellon, (2011) suggested that “the essence of olfaction” is to provide a detailed representation of the complex chemical environment integrating chemical signals from a variety of interesting sources (…) without reference to the location of stimuli (…). In contrast, the essence of “distributed chemoreception” is to form representations of only few key chemicals (food-related chemicals, pheromones) within a somatotopic context provided by mechanoreception. The integration of chemo- and mechanosensory information permits pinpointing the location of chemical stimuli …”

Independently of insects, chelicerates, and myriapods, terrestrial Crustacea provide a fascinating chance to look on a wonderful evolutionary experiment by analyzing which potential alternative solutions arthropods have evolved to explore the terrestrial olfactory landscape (Hansson et al., 2011). We have previously analyzed the olfactory system of land hermit crabs (Anomala, Coenobitidae) including their peripheral (Stensmyr et al., 2005; Tuchina et al., 2014) and central olfactory pathway (Harzsch & Hansson, 2008; Krieger et al., 2010; Polanska et al., 2012; Wolff et al., 2012; Tuchina et al., 2015), in addition to behavioral and physiological aspects (Stensmyr et al., 2005; Krång et al., 2012). These studies provided evidence for coenobitids having a superb sense of aerial olfaction. In this paper, we ask whether terrestrial brachyuran crabs also evolved the neuronal basis for aerial olfaction. Therefore, we compare the anatomy of the central olfactory pathway of selected species of brachyuran crustaceans featuring a rather terrestrial lifestyle to that of their marine relatives.

Material and methods

Experimental animals

We analyzed representatives of several different species of brachyurans representing aquatic species (four exclusively marine and one freshwater crab species) as well as four brachyuran species featuring different grades of terrestrial adaptation (Table 1; Figs. 2 and 3). For simplification, the four latter brachyurans are referred as terrestrial brachyurans throughout this text, although all nine species feature terrestrial adaptions at various degrees (Table 1). After shipping, living specimens of Cardisoma armatum, Geosesarma tiomanicum, and Uca tangeri were kept in tanks providing both a water and a land part. Husbandry as well as observation and documentation of these species were conducted between 5 and 14 days until dissection in the laboratory. Before dissection, animals were sexed, and the carapace width as well as the wet weight of each animal was measured. The collection of specimens of Gecarcoidea natalis was permitted by Christmas Island National Park (Australian Government; Department of the Environment; Parks Australia; Permit No.: AU_COM 2010-090-1).

Table 1 Investigated species including the sexes, numbers, grades of terrestriality (T1–T5, where T1 is the lowest grade and T5 the highest), and origins of specimens.

Species	Taxon	Sex (n ind.)	Grade(s) of terrestrial adaptation	Source/origin;	
Cardisoma armatum, Herklots, 1851	Gecarcinidae	♀(5)	T 3	https://www.interaquaristik.de	
Gecarcoidea natalis, (Pocock, 1888)	Gecarcinidae	♂(2); ♀(2)	T 4	Christmas Island (Australia)	
Geosesarma tiomanicum, Ng, 1986	Sesarmidae	♂(2); ♀(2)	T 5	https://www.interaquaristik.de	
Uca tangeri, (Eydoux, 1935)	Ocypodidae	♂(2); ♀(2)	T2–T3	www.tropicwater.eu	
Epilobocera sinuatifrons, Rathbun 1866	Pseudothelphusidae	n.a. (4)	T2–T3	Guajataca, Puerto Rico	
Carcinus maenas, (Linnaeus, 1758)	Portunidae	♂(4); ♀(11)	T 1	Marine Science Center in Rostock (Germany)	
Percnon gibbesi, (Milne-Edwards, 1853)	Percnidae	♂(1); ♀(1)	T 1	Mediterranean, Cala Llenya (Ibiza, Spain)	
Xantho hydrophilus, (Herbst, 1790)	Xanthidae	♂(4); ♀(2)	T 1	Mediterranean, Cala Llenya (Ibiza, Spain)	
Xantho poressa, (Olivi, 1792)	Xanthidae	♂(2); ♀(2)	T 1	Mediterranean, Cala Llenya (Ibiza, Spain)	
Pagurus bernhardus (Linnaeus, 1758)	Paguridae	n.a. (1)	T 1	North Atlantic Ocean, Roscoff (France)	

Figure 2 Brachyuran species analyzed.

Portraits of individuals of investigated brachyuran species in living state. (A–C) Marine Brachyura: (A) Carcinus maenas (Boiensdorf, Baltic Sea, 1998), (B) Percnon gibbesi (infralittoral rock bottom, 1 m depth, Cala Llenya, Ibiza, Spain, 2013), (C) Xantho hydrophilus (infralittoral rock bottom, 5 m depth, Cala Llenya, Spain, 2012). (D–H) Terrestrial Brachyura: (D) Epilobocera sinuatifrons (Guajataca, Puerto Rico, 2004), (E) Gecarcoidea natalis (At the Pink House, Christmas Island, Australia, 2011), (F) Cardisoma armatum, (G) Geosesarma tiomanicum, (H) Uca tangeri. Portraits of (F–H) in living state were taken in the laboratory in Greifswald in 2013.

Figure 3 Comparative draft of studied animals: their brains, first antennae, deutocerebral chemosensory lobes (DCLs) and olfactory glomeruli.

Note that drawings are equally scaled in each line. Each column corresponds to same species as follows: Carcinus maenas, Percnon gibbesi, Xantho hydrophilus, as representatives for marine brachyurans are given and opposed (separated by a dashed line) to Gecarcoidea natalis, Cardisoma armatum, Geosesarma tiomanicum, and Uca tangeri representing brachyuran species featuring terrestrial lifestyles to different degrees. Note that for animals featuring a markedly size-specific sexual dimorphism, only the males are drawn. (A) Dorsal view of habitus in all studied species. (B) Distal antennomeres of the first antennae (antennules) of all species featuring the minor median and major lateral flagella which bear the aeasthetascs. (C) Outlines of central brains based on the synapsin immunoreactivity. The lateral protocerebrum and nerves are not displayed. (D) Outlines of DCLs and peripheral arrangement of olfactory glomeruli as they appear in horizontal sections. Note that the position of DCL within the brain is indicated in C. maenas in line C. (E) Examples of shape and organization of randomly chosen olfactory glomeruli of all studied species as they appear in horizontal sections.

Analysis of antennae and aesthetascs

The first pairs (antennules) and the second pairs (antennae) of post-ocular appendages were cut off prior to brain dissection and were transferred into 70% ethanol (G. natalis and two animals of G. tiomanicum) or in 2% glutaraldehyde in 0.1 M phosphate buffered saline (PBS; further specimens of G. tiomanicum, C. armatum and U. tangeri). Micrographs of these appendages were documented in PBS with a Nikon Eclipse 90i microscope equipped with a digital camera (Nikon DS2-MBWc) and analyzed by the use of the software package NIS-Elements AR. Cuticular auto-fluorescence of the first and second antennae was excited with ultraviolet light (UV) with a wavelength of 340–380 nm eliciting light emissions with a wavelength of 435–485 nm. Aesthetascs of marine specimens such as of Pagurus bernhardus (Linnaeus, 1758), Carcinus maenas, Xantho hydrophilus, X. poressa and Percnon gibbesi were cut off from the lateral flagella with a razor blade and counted on an object slide using UV-excitation as well as bright field illumination.

Histochemistry, immunohistochemistry, and microscopy

The animals were anaesthertized by cooling on ice for 1 h before dissection. Following the protocol by Ott (2008), the dissected brains were fixed in toto for approximately 20 h (room temperature) in 3.7% formaldehyde/zinc-fixative (the ready-to-use formaldehyde/zinc-fixative was obtained via Electron Microscopy Sciences. Cat. No. 15675). For whole-mount preparations, the brains and eyestalk ganglia were dissected, and the retina including all pigments was removed. The whole-mounts were washed three times in HEPES-buffered saline (HBS) for 15 min, subsequently transferred to Dent’s fixative (80% methanol/20% DMSO), and post-fixated for two hours at room temperature. Specimens were then transferred to 100% methanol and stored overnight in the refrigerator and rehydrated stepwise for 10 min each in 90%, 70%, 50%, 30% methanol in 0.1 M Tris–HCl-buffer, and finally in pure 0.1 M Tris–HCl-buffer (pH 7.4). Alternatively, for preparing horizontal brain sections, after anaesthetizing the animals by cooling on ice for 1 h, the brains were dissected and fixed in 4% paraformaldehyde (PFA) in 0.1 M PBS overnight. The dissected brains were washed for 4 h in several changes of PBS and sectioned (80–100 µm sections) horizontally at room temperature using a vibratome (Zeiss Hyrax V590; Carl Zeiss, Oberkochen, Germany). For permeation of cell membranes, both brain whole-mounts as well as brain sections were then preincubated for 90 min in PBS-TX (1% Bovine-Serum-Albumine, 0.3% TritonX-100, 0.05% Na-acide, in 0.1 M PBS; pH 7.4). In contrast to the protocol of Ott (2008), PBS-TX was used instead of PBSd-NGS. Finally, the samples were incubated at 4 °C for 84 h (whole-mounts) or overnight (sections) in the primary antisera. The following sets of reagents were used (compare Krieger et al., 2012):

Set A: rabbit anti-Dip-allatostatin 1 (AST-A; final dilution 1:2,000 in PBS-TX; antibody provided by H Agricola, Friedrich-Schiller Universität Jena, Germany); monoclonal mouse anti-synapsin “SYNORF1” antibody (final dilution 1:30 in PBS-TX; antibody provided by E Buchner, Universität Würzburg, Germany) detected by anti- mouse Cy3 (CyTM3-conjugated AffiniPure Goat Anti-Mouse IgG Antibody; Jackson ImmunoResearch Laboratories Inc., West Grove, PA, USA).

Set B: polyclonal rabbit anti-FMRFamid (in PBS-TX; final dilution 1:2,000; Acris/Immunostar; Cat. No. 20091) detected by anti-rabbit Alexa Flour 488 (IgG Antibody, invitrogen, Molecular Probes); monoclonal mouse anti-synapsin “SYNORF1” antibody (in PBS-TX; final dilution 1:30; antibody provided by E Buchner, Universität Würzburg, Germany) detected by anti- mouse Cy3 (CyTM3-conjugated AffiniPure Goat Anti-Mouse IgG Antibody; Jackson ImmunoResearch Laboratories Inc., West Grove, PA, USA).

Set C: monoclonal mouse anti-synapsin “SYNORF1” antibody (in PBS-TX; final dilution 1:30; antibody provided by E Buchner, Universität Würzburg, Germany) detected by anti- mouse Alexa Flour 488 (IgG Antibody; Invitrogen, Waltham, MA, USA; Molecular Probes, Eugene, OR, USA); counterstain: phallotoxins conjugated to Alexa Fluor 546 (concentration 200 units/ml; Molecular Probes, Eugene, OR, USA) as a high-affinity probe for f-actin.

In all three sets, the tissues were incubated in mixture containing the secondary antisera and the nuclear marker HOECHST (33,242; 0.1 µg/ml) for 2.5 days at 4 °C (whole-mounts) or for 4 h at room temperature (sections). Finally, the brain sections were washed for at least 2 h in several changes of PBS at room temperature and mounted in Mowiol® (Calbiochem) between two coverslips. After secondary antibody incubation, the whole-mounts were dehydrated in changes of ascending glycerol concentrations (1%, 2%, 4% (2 h each), 8%, 15%, 30%, 50%, 60%, 70%, and 80% (1 h each) glycerol diluted in Tris–HCl buffer, with DMSO to 1% final concentration). After the last step of dehydration, the whole-mounts were washed twice for 30 min in 99.6% denatured ethanol. The ethanol was then underlyed by the same volume of methylsalicylate for clearing of the whole-mount brains. After the brains were cleared, the supernatant liquid was removed and the samples were and then mounted in customized chambers (a custom washer from the hardware store was glued between two coverslips as spacer) filled with methylsalicylate and sealed with Mowiol®. The triple-labeled and sectioned tissues were analyzed using a Nikon eclipse 90i microscope equipped with a digital camera (Nikon DS2-MBWc). The whole-mounts were analyzed by using a confocal laser scanning microscope (clsm; Leica TCS SP5II; Leica, Wetzlar, Germany). The pictures were then processed using the NIS-Elements AR software and Adobe Photoshop CS4. Only global picture enhancement features of Photoshop elements (black to white inversion, brightness, and contrast) were used for all experiments. Three-dimensional (3D) brain reconstructions in addition to volumetric analysis based on optical section series of clsm data were performed using the reconstruction software Amira® (FEI Visualization Sciences Group, Mérignac, France).

Raw data of brain section series is available from https://www.morphdbase.de under the “media” tab under a combination of the short title “Krieger” and an identifier according to the species and ID of the specimen.

Three-dimensional reconstructions of brains and substructures are based on tomographies of three specimens per species for C. armatum, G. tiomanicum and U. tangeri. For each specimen, surfaces of one DCL including the corresponding olfactory glomeruli and the ipsilateral AcN were generated by manual labeling. Finally, the computed 3D surfaces were slightly smoothed and resulting parameters such as the glomerular number and volume as well as the volume of the whole DCL were analyzed.

Antibody specificity

Synapsin

The monoclonal mouse anti-Drosophila synapsin “SYNORF1” antibody (provided by E Buchner, Universität Würzburg, Germany) was raised against a Drosophila GST-synapsin fusion protein and recognizes at least four synapsin isoforms (ca. 70, 74, 80, and 143 kDa) in western blots of Drosophila head homogenates (Klagges et al., 1996). In western blot analysis of crayfish homogenates, this antibody stains a single band at ca. 75 kDa (see Sullivan et al., 2007). Harzsch & Hansson (2008) conducted western blot analysis comparing brain tissue of Drosophila and the hermit crab Coenobita clypeatus which is closely related to the species studied in this contribution. The antibody provided identical results for both species, staining one strong band around 80–90 kDa and a second weaker band slightly above 148 kDa (see Harzsch & Hansson, 2008). Their analysis strongly suggests that the epitope which SYNORF 1 recognizes is strongly conserved between the fruit fly and the hermit crab. Similar to Drosophila, the antibody consistently labels brain structures in representatives of all major subgroups of the malacostracan crustaceans (see Beltz et al., 2003; Harzsch, Anger & Dawirs, 1997; Harzsch et al., 1998; Harzsch et al., 1999; Harzsch & Hansson, 2008; Vilpoux, Sandeman & Harzsch, 2006; Krieger et al., 2010; Krieger et al., 2012) in a pattern that is consistent with the assumption that this antibody labels synaptic neuropils in Crustacea. In the crustacean first visual neuropil (the lamina), synapsin labeling is weak compared to the other brain neuropils (Harzsch, Anger & Dawirs, 1997; Harzsch & Hansson, 2008). Similarly, in Drosophila melanogaster labeling of the lamina is weak, because photoreceptors R1–R6 which have their synapses in the lamina contain very little of the presently known synapsin isoforms (Klagges et al., 1996). The antibody also labels neuromuscular synapses both in Drosophila and in Crustacea (Harzsch, Anger & Dawirs, 1997). These close parallels in the labeling pattern of SYNORF1 between Drosophila and various Crustacea strengthen the claim that it also recognizes crustacean synapsin homologues. This antibody even labels synaptic neuropil in an ancestral clade of protostomes, the Chaetognatha (Harzsch & Müller, 2007) suggesting that the epitope recognized by this antiserum is conserved over wide evolutionary distances.

Allatostatin A

The A-type allatostatins (A-ASTs; synonym dip-allatostatins) constitute a large family of neuropeptides that were first identified from the cockroach Diploptera punctata and that share the C-terminal motif-YXFGLamide (reviews Stay, Tobe & Bendena, 1995; Nässel & Homberg, 2006; Stay & Tobe, 2007). In decapod crustaceans, almost 20 native A-ASTs and related peptides were initially identified from extracts of the thoracic ganglia of the shore crab Carcinus maenas (Duve et al., 1997), and shortly after several other A-ASTs were isolated from the freshwater crayfish Orconectes limosus (Dircksen et al., 1999). Meanwhile, the family of crustacean A-ASTs has substantially grown to several dozens of representatives (review Christie, Stemmler & Dickinson, 2010) with additional members being discovered in the prawns Penaeus monodon (Duve et al., 2002) and Macrobrachium rosenbergii (Yin et al., 2006), in the brachyuran crabs Cancer borealis (Huybrechts et al., 2003) and Cancer productus (Fu, Christie & Li, 2005), Carcinus maenas (Ma et al., 2009a), the crayfish Procambarus clarkii (Yasuda-Kamatani & Yasuda, 2006), the lobster Homarus americanus (Cape et al., 2008; Ma et al., 2008; Ma et al., 2009b) the shrimps Litopenaeus vannamei (Ma et al., 2010) as well as a non-malacostracan crustacean, the copepod Calanus finmarchicus (Christie et al., 2008).

We used an antiserum that was raised against the Diploptera punctata (Pacific beetle cockroach) A-type Dip-allatostatin I, APSGAQRLYGFGLamide, coupled to bovine thyroglobulin using glutaraldehyde (Vitzthum, Homberg & Agricola, 1996) that was kindly provided by H Agricola (Friedrich-Schiller Universität Jena, Germany) and that previously has been used to localize A-ASTs in crustacean and insect nervous systems (e.g., Vitzthum, Homberg & Agricola, 1996; Dircksen et al., 1999; Skiebe, 1999; Utting et al., 2000; Kreissl, Strasser & Galizia, 2010). Competitive ELISA with DIP-allatostatin I, II, III, IV and B2 showed that the antiserum is two orders of magnitude more sensitive to Dip-allatostatin I than to Dip-allatostatins II, III, IV, and B2 (Vitzthum, Homberg & Agricola, 1996). Vitzthum, Homberg & Agricola (1996) have reported that the antiserum displays no cross-reactivity with corazonin, CCAP, FMRFamide, leucomyosuppression, locustatachykinin 11, perisulfakinin, and proctolin as tested by non-competitive ELISA. Preadsorption of the diluted antisera against Dip-allatostatin I, GMAP and Manduca sexta allatotropin with 10 µM of their respective antigens abolished all immunostaining in brain sections of Schistocerca gregaria (Vitzthum, Homberg & Agricola, 1996). A sensitive competitive enzyme immunoassay (EIA) confirmed the high specificity of the antiserum for A-type Dip-allatostatin I (Dircksen et al., 1999). In the brains of the honey bee Apis mellifera, preadsorption controls with AST I and AST VI completely abolished all staining of the antiserum (Kreissl, Strasser & Galizia, 2010). Sombke, Harzsch & Hansson (2011) repeated a preadsorption test in Scutigera coleoptrata and preincubated the antiserum with 200 µg/ml A-type allatostatin I (A9929; 16 h 4 °C; Sigma-Aldrich, St. Louis, MO, USA) and this preincubation abolished all staining. Preadsorption of the antiserum with AST-3 was reported to abolish all labeling in the stomatogastric nervous system of the crab Cancer pagurus, the lobster Homarus americanus and the crayfish Cherax destructor and Procambarus clarki (Skiebe, 1999). It seems safe to assume that this antiserum most likely binds to all A-ASTs that share a -YXFGLamide core. However, the term “allatostatin-like immunoreactivity” is used throughout this work, because it may possible that the antibody also binds related peptides.

RFamide-related peptides

The tetrapeptide FMRFamide and FMRFamide-related peptides (FaRPs) are prevalent among invertebrates and vertebrates and form a large neuropeptide family with more than 50 members all of which share the RFamide motif (Price & Greenberg, 1989; Greenberg & Price, 1992; Nässel, 1993; Homberg, 1994; Dockray, 2004; Nässel & Homberg, 2006; Zajac & Mollereau, 2006). In malacostracan Crustacea, at least twelve FaRPs have been identified and sequenced from crabs, shrimps, lobsters, and crayfish (Huybrechts et al., 2003; Mercier, Friedrich & Boldt, 2003), which range from seven to twelve amino acids in length and most of them share the carboxy-terminal sequence Leu-Arg-Phe-amide. The utilized antiserum was generated in rabbit against synthetic FMRFamide (Phe-Met-Arg-Phe-amide) conjugated to bovine thyroglobulin (DiaSorin, Cat. No. 20091, Lot No. 923602). According to the manufacturer, immunohistochemistry with this antiserum are completely eliminated by pretreatment of the diluted antibody with 100 µg/ml of FMRFamide. Harzsch & Hansson (2008) repeated this experiment in the anomalan Coenobita clypeatus which is closely related to the species studied here, specifically to the hermit crabs, and preincubated the antiserum with 100 µg/ml FMRFamide (16 h, 4 °C; Sigma-Aldrich, St. Louis, MO, USA) resulting in a complete abolishment of all staining. Because the crustacean FaRPs know so far all share the carboxy-terminal sequence LRFamide, we conclude that the DiaSorin antiserum that we used most likely labels any peptide terminating with the sequence RFamide. Therefore, we will refer to the labeled structures in our specimens as “RFamide-like immunoreactivity” throughout the paper.

Nomenclature

The neuroanatomical nomenclature used in this manuscript is based on Sandeman et al. (1992) and Richter et al. (2010) with some modifications adopted from Harzsch & Hansson (2008)) and Loesel et al. (2013). In favor of a consistent terminology, here we suggest avoiding the term “optic neuropils” (Hanström, 1925; Sombke & Harzsch, 2015) as well as “optic lobes” (Kenyon, 1896). Even if the Greek “optikos” and the Latin term “visus” have the identical meaning, nowadays, “optic” in the field of visional anatomy and physiology refers to the physically refractive components of the eye for the reception of light. To emphasize the perceptive character of these neuropils, we suggest using the term “visual neuropils” which is consistent with, e.g., the visual cortex in mammals, formerly also termed “optic” cortex (Spiller, 1898). All post-retinal components that are related to vision, such as the “optic tract” and the “inner” as well as the “outer optic chiasm” should be consequently renamed, too. Here, we suggest to use “visual tract” (VT) and the “inner” (iCh) as well as the “outer visual chiasm” (oCh) accordingly. However, for all pre-retinal components that are related to vision, the term “optic,” as for example in the dioptric apparatus of the ommatidia, should be maintained. We also discourage the commonly used terms “eyestalk neuropils” (Bliss & Welsh, 1952; Polanska, Yasuda & Harzsch, 2007), “optic ganglia” (Medan et al., 2015), or “eyestalk ganglia” (Harzsch & Dawirs, 1996; Techa & Chung, 2015), usually summarizing the visual neuropils as well as the neuropils of the TM/HN-complex, because these neuropils together can be located more proximal to the central brain and not in the eyestalk in some species, and thus are part of the central brain as exemplified below. Furthermore, the neuropils of the lateral protocerebrum (visual neuropils + TM/HN-complex) do not fulfill the definition of a ganglion (see Richter et al., 2010). The traditional nomenclature of the visual neuropils lamina ganglionaris, medulla interna, and medulla externa has been modified as suggested by Harzsch (2002) to lamina, medulla, and lobula. Because we could not detect any border between the cell body clusters (9) and (11) of olfactory interneurons as described in Sandeman et al. (1992), we collectively refer to them as cluster (9/11) (see Krieger et al. 2010). The term “oesophageal connective” and the corresponding abbreviation OC (British English) are maintained here for simplicity. The olfactory neuropil (ON or OL) is now named the deutocerebral chemosensory lobe (DCL), and the olfactory globular tract (OGT) is now named the projection neuron tract (PNT) according to Loesel et al. (2013). Consequently, the olfactory globular tract neuropil OGTN is now named projection neuron tract neuropil (PNTN). For simplification, the neuroanatomical descriptions are kept restricted to only one hemisphere of the brain and hold true for all specimens studied if not stated otherwise.

The data presented in this study are drawn from different sets of triple-labeling immunofluorescence experiments as laid out above. The localizations of synapsins provides a general labeling of all neuropils in the brain whereas staining of actin is better suited to label neurite bundles and fiber tracts. The two antisera against allatostatin and FMRFamide label specific neuronal subsets and were chosen for a better comparison with other studies that have used the same markers (e.g., Harzsch & Hansson, 2008; Krieger et al., 2010; Krieger et al., 2012). The following abbreviations (color-coded in the figures) identify the markers:

SYN synapsin-like immunoreactivity (magenta or black)

RFA RFamid-like immunoreactivity (green or black)

PHA actin labeling by the use of phalloidin (green or black)

AST allatostatin-like immunoreactivity (green or black)

NUC nuclear counterstain with HOECHST-dye H 33258 (cyan or black)

Results

The antennae

In general, the first antennae in brachyuran crustaceans each consists of two branches called the median and the lateral flagellum (Fig. 4). Both flagella are composed of several units, the flagellomeres. Each flagellomere of the lateral flagellum is equipped with one row of the typical unimodal chemosensory sensilla, the aesthetascs, in both marine and terrestrial brachyurans (Fig. 5). A quantification of aesthetasc numbers is provided in Table 2. The shape of the aesthetascs in marine versus terrestrial brachyurans displays marked differences. In the marine species, the aesthetascs are long and slender, whereas in all species featuring a rather terrestrial lifestyle, they are short and blunt (Fig. 5). The second antennae consist of one articulated branch only, composed of multiple antennomeres, and with the low-resolution light microscopic methods used here, we could not detect any striking differences in the sensillar equipment between the marine and terrestrial representatives (Fig. 6).

Figure 4 First antenna in studied brachyuran species.

(A) UV-autofluorescence micrograph shows equally scaled first antenna (AI) from four marine species Carcinus maenas, Percnon gibbesi, Xantho hydrophilus and Xantho poressa; and (B) from four terrestrial species Gecarcoidea natalis, Cardisoma armatum, Geosesarma tiomanicum and Uca tangeri. Abbreviations: AS, aesthetascs; lFl, lateral flagellum; mFl, median flagellum.

Figure 5 Flagella and aesthetascs on first antenna in different brachyuran species.

(A) UV-autofluorescence micrograph shows lateral and median flagellum as well as the aesthetascs from four marine species: Carcinus maenas, Percnon gibbesi, Xantho hydrophilus, and Xantho poressa and (B) from four terrestrial species Cardisoma armatum, Gecarcoidea natalis, Geosesarma tiomanicum, and Uca tangeri. A micrograph using transmitted light shows the lateral flagellum and aesthetascs from Geosesarma tiomanicum. Asterisks identify single annuli of the lateral flagellum. Abbreviations: AS, aesthetascs; lFl, lateral flagellum; mFl, median flagellum.

Figure 6 Second antenna of studied brachyuran species.

(A) UV-autofluorescence micrograph shows the equally scaled second antenna (AII) from four marine species Carcinus maenas, Percnon gibbesi, Xantho hydrophilus and Xantho poressa and (B) from four terrestrial species Gecarcoidea natalis, Cardisoma armatum, Geosesarma tiomanicum and Uca tangeri.

Table 2 Morphometric data of structures within the peripheral olfactory pathway and of the primary olfactory centers in the brain of decapod crustaceans.

Taxon	Species	n ind.	Aesth. number	Aesth. length (μm)	DCL volume (103μm3)	Glom. volume (103μm3)	Glom. number	Reference	
Achelata	Panulirus interruptus	2	1,786	–	344,922	288	1,202	Beltz et al. (2003)	
	Panulirus argus	2	1,255	–	154,069	118	1,332	Beltz et al. (2003)	
	Panulirus argus	–	–	–	–	–	≈750	Blaustein et al. (1988)	
	Panulirus argus	–	–	–	–	–	≈1,100	Schmidt & Ache (1997)	
	Jasus edwardsii	3	1,537	–	591,956	616	961	Beltz et al. (2003)	
Homarida	Homarus americanus	2	1,262	–	141,160	592	249	Beltz et al. (2003)	
	Homarus americanus	–	–	–	–	–	90–200	Helluy et al. (1996)	
Astacida	Cherax destructor	3	130	–	24,187	111	230	Beltz et al. (2003)	
	Cherax destructor	–	–	–	–	–	≈100	Sandeman & Luff (1973)	
	Cherax quadricarinatus	3	237	–	24,736	74	334	Beltz et al. (2003)	
	Procambarus clarkii	3	133	–	9,790	20	503	Beltz et al. (2003)	
	Procambarus clarkii	–	–	–	–	–	≈150	Blaustein et al. (1988)	
	Procambarus clarkii	–	–	–	–	–	20–200	Mellon Jr & Alones (1993)	
Thalassinida	Callianassa australiensis	3	22		6,589	28	235	Beltz et al. (2003)	
Anomala	Pagurus bernhardus	1	736	≈1,200	–	170	–	Krieger et al. (2012)	
	Pagurus bernhardus	–	673	–	–	–	536	Tuchina et al. (2015)	
	Coenobita clypeatus	3 (*1)	519	*80–100	120,352	154	799	Beltz et al. (2003)	
	Birgus latro	1 (*2)	1700	*100–200	374,682	280	1,338	Krieger et al. (2010)	
	Birgus latro	1	780	–	–	–	–	Harms (1932)	
	Petrolisthes coccnicus	3	328	–	12,359	19	655	Beltz et al. (2003)	
Brachyura	Cancer borealis	2	540	–	165,731	230	733	Beltz et al. (2003)	
	Carcinus maenas	1	285	≈700	–	230	–	Krieger et al. (2012)	
	Xantho hydrophilus	2	206	≈750	–	–	–	This paper	
	Xantho poressa	2	222	≈600	–	–	–	This paper	
	Libinia dubia	3	319	–	20,327	39	454	Beltz et al. (2003)	
	Percnon planissimum	3	555	–	28,765	59	495	Beltz et al. (2003)	
	Percnon gibbesi	2	165	≈700	–	–	–	This paper	
	Paragrapsus gaimardii	–	160–170	600	–	–	–	Snow (1973)	
	Cardisoma armatum	3 (*5)	*84	*125–−150	12,605	74	69	This paper	
	Sesarma sp.	3	33	–	6,617	15	446	Beltz et al. (2003)	
	Gecarcoidea natalis	1 (*3)	*113	*100–125	9,432	49	193	This paper	
	Geosesarma tiomanicum	3 (*5)	*26	*60–80	4,253	21	61	This paper	
	Uca tangeri(♂+♀)	3 (*6)	*38	*90–110	5,355	42 ± 29	64	This paper	
	Uca tangeri (♀)	2 (*3)	*36		5,300	20 ± 6	78	This paper	
	Uca tangeri (♂)	1 (*3)	*40		5,400	86	36	This paper	
	Uca minax	3	39	–	4,558	18	284	Beltz et al. (2003)	
	Uca pugilator	3	28	–	3,115	13	234	Beltz et al. (2003)	
	Uca pugnax	3	26	–	3,012	8	374	Beltz et al. (2003)	
Note that volumes of DCLs and olfactory glomeruli (glom.) are estimates based on a variety of different neuroanatomical methods (for further information see references). All volumes are averaged for one single structure (one DCL per hemisphere or one average glomerulus), rounded to the nearest 1,000 µm3 and are thus slightly modified from the original literature. Note that for each individual investigated, the number of aesthetascs (aesth.) per antenna are based on one randomly chosen antenna per pair. The table is compiled after Beltz et al. (2003), Schachtner, Schmidt & Homberg (2005) and Krieger et al. (2010) and complemented with data of other authors (see reference column) as well as with our own data (in bold) in addition of aesthetasc lengths. Note that the aesthetasc lengths in B. latro (upper range estimated from Stensmyr et al. (2005) and lower range from J Krieger, 2010, unpublished data) and C. clypeatus (unpublished data) are estimated based on scanning electron micrographs. Associated subsets of morphometric data apart from the main data set are indicated by asterisks.

The brain

General arrangement of neuropils in the brachyuran brain

The general morphology of the brachyuran brain, as in other Malacostraca, is composed of three consecutive neuromeres, the proto-, deuto-, and tritocerebrum as extensively reported in previous studies (reviewed in Harzsch, Sandeman & Chaigneau, 2012; Schmidt, in press). In some anomalan species, such as Birgus latro or Petrolisthes lamarckii as well as in the axiid shrimp Callianassa australiensis, the bilaterally paired visual neuropils and the neuropils of the terminal medulla/hemiellipsoid body—complex (TM/HN-complex) are located anteriorly adjacent to the “central” brain as a consequence of elongated axons composing the optic nerve. In all brachyurans studied so far, however, these neuropils are located within the eyestalks, thus being situated at some distance from the central portion of the syncerebrum. Note that in the comparative Fig. 3, for simplicity, only outlines of the central portions of the brains, in the following simply termed the “central brain”—are drawn, without the neuropils of the lateral protocerebrum. In horizontal sections, this central brain appears broader than elongated along the neuraxis (Fig. 3C). The species studied here displayed markedly different carapace widths ranging from 14 mm in G. tiomanicum up to 90 mm in G. natalis. In contrast, the general brain dimensions are rather similar across species as indicated by a range of brain width between 1.4 mm in G. tiomanicum to 2.5 mm in G. natalis and 2.7 mm in C. maenas. Hence, there seems to be only a weak correlation between brain size and body size.

Contrary to most other decapods analyzed so far (see e.g., Sandeman, Scholtz & Sandeman, 1993), a distinct compartmentalization of the brain neuropils is less obvious in brachyurans. For instance, the neuropil boundaries in true crabs are much less distinct than in Anomala (compare Krieger et al., 2012). However, the general organization of the brachyuran brain and arrangement of its subunits can be deduced from anatomical data by tracing nerves as well as interconnecting tracts between the corresponding neuropils as outlined below:

The protocerebral tract (PT) is composed of neurites originating in neuropils of the lateral protocerebrum (lPC). The PT interconnects these neuropils with the proximal part of the brain, the median protocerebrum (mPC). The median protocerebrum is composed of the anterior (AMPN) and the posterior medial protocerebral neuropil (PMPN), which together resemble the shape of a butterfly in horizontal brain sections (compare Figs. 3, 9D, 9E, 11D, 11E, 13C–13G, 15A–15D, 16A1–16C1 and 17). Both neuropils are almost completely fused anterioposteriorly as well as across the midline with their contralateral counterparts into one single neuropil mass in the brachyuran brain, but they appear separated in horizontal sections at the level of the central body (Figs. 11E–11F, 13D, 15B and 15F). Furthermore, the median protocerebrum includes neuropils of the central complex, namely from anterior to posterior: the unpaired protocerebral bridge (PB), the unpaired central body (CB), and the bilaterally paired lateral accessory lobes (Lals).

In all brachyuran species investigated, the neuropils of the deutocerebrum (DC) that extend posteriorly adjacent to the median protocerebrum consist of the unpaired median antenna I neuropil (MAN), the bilaterally paired antenna I neuropils (LANs), the deutocerebral chemosensory lobes (DCLs; formerly referred as olfactory lobes or olfactory neuropils), the accessory lobes (AcNs) and the projection neuron tract neuropils (PNTNs; formerly referred as olfactory globular tract neuropils or OGTNs). Each DCL consists of several to hundreds of barrel-shaped subunits of synaptic neuropil, the olfactory glomeruli (OG) which are arranged in a radial, palisade-like array in the periphery of the lobe. Medially to each DCL, a cluster of somata (9/11) of hundreds of interneurons of varying sizes is present. These neurons extend neurites which enter the DCL via the median foramen (mF), one of three gaps in the palisade-like array of olfactory glomeruli. Furthermore, several hundreds of somata of olfactory projection neurons are grouped in cell cluster (10) posteriorly to each DCL. Their neurites enter each DCL via the posterior foramen (pF), innervate the olfactory glomeruli, and project axons that exit each lobe via its median foramen (mF) in a large bundle that constitutes the projection neuron tract (PNT). The axons of the PNT interconnect each DCL with the ipsilateral as well as contralateral hemiellipsoid body within the lateral protocerebrum by forming a chiasm dorsally of the central body.

The tritocerebrum (TC) posteriorly adjoins the neuropils of the deutocerebrum and is composed of the bilaterally paired antenna II neuropils (AnNs) and further dorsally, of the tegumentary neuropils (TNs).

Figure 7 Optical horizontal sections of lateral protocerebrum and central brain in Cardisoma armatum.

(A–E) Micrographs of triple-labeled optical horizontal sections showing visual neuropils and the lateral protocerebrum. Lamina was lost through dissection. (F–J) brain and details of specific brain areas such as, median protocerebrum and deutocerebrum in H and protocerebral bridge (PB) in I. The arrow with a dashed line marks a giant neuron in I. Note that (B, D, E, G and J) show inverted single-channel micrographs of different labelings (indicated by abbreviations). Abbreviations of immunhistochemical labelings and histochemical markers: NUC, nuclear marker (cyan); PHA, actin-labeling using phalloidin (green or black); RFA, labeling against RFamide (black); SYN, labeling against synapsin (magenta or black). Other abbreviations: 2, 3, 4/5, 6, and 9/11, cell clusters (2), (3), (4/5), (6), and (9/11); AINv, antenna I nerve; AcN, accessory neuropil; AMPN, anterior medial protocerebral neuropil; CB, central body, DCL, deutocerebral chemosensory lobe; HN, hemiellipsoid body; iCh, inner visual chiasm; LAN, lateral antenna I neuropil; Lo, lobula; MAN, median antenna I neuropil; Me, medulla; oCh, outer visual chiasm; PMPN, posterior medial protocerebral neuropil; PNT, projection neuron tract; PT, protocerebral tract; TM, terminal medulla; VT, visual tract. Scale bars, 250 µm.

Lateral protocerebrum: the visual neuropils and the terminal medulla/hemiellipsoid body—complex (TM/HN-complex)

The eyestalks of most malacostracan crustaceans each contain three successive retinotopic neuropils. These three main visual neuropils process visual input and from distal to proximal are termed the lamina, medulla, and lobula. An additional (fourth) neuropil can be found adjacent to the lobula referred to as lobula plate. If present, the lobula plate adheres the lobula. The architecture of these visual neuropils which often are referred to as optic neuropils is best known in crayfish and lobsters (review Harzsch, Sandeman & Chaigneau, 2012) but was also analyzed in a number of marine and amphibious brachyurans including Chasmagnathus granulatus, Hemigrapsus oregonensis (Sztarker, Strausfeld & Tomsic, 2005; Sztarker et al., 2009; Berón de Astrada, Medan & Tomsic, 2011; Berón de Astrada et al., 2013), and Carcinus maenas (Elofsson & Hagberg, 1986; Krieger et al., 2012). Although the visual neuropils are not the focus of the present study, successful eyestalk preparations from C. armatum (Figs. 7A–7D), G. natalis (Figs. 9A–9C), G. tiomanicum (Figs. 11A and 11B), and U. tangeri (Fig. 13A) show that these terrestrial species display well developed visual neuropils that show distinct synapsin-like immunoreactivity (SYN). Distinct clusters of somata become clearly visible distal to each visual neuropil. According to their appearance from distal to proximal and based on nuclear counterstaining, we distinguish cluster (1) associated with the lamina; cluster (2) associated with the medulla; cluster (3) associated with the lobula (Figs. 7A, 9A, 11A and 13A). Their arrangement and layered architecture closely correspond to those of their marine relatives. In the lobula, we could resolve three main layers in all four species (Figs. 7B, 9B, 11A and 13A) suggesting that at the level of resolution we analyzed, the visual neuropils show a high level of similarity.

The most proximal neuropils of the lateral protocerebrum, the terminal medulla (TM; also termed medulla terminalis) and the hemiellipsoid body (HN), which are considered multimodal associative areas (Wolff et al., 2012), are located within the eyestalk and together constitute an almost spherical neuropil mass (TM/HN-complex) in the species studied. They are identifiable in preparations of C. armatum (Figs. 7A–7E), G. natalis (Figs. 9A–9C), G. tiomanicum (Figs. 11A–11C2), and U. tangeri (Figs. 13A and 13B) showing distinct SYN, but were also described in Chasmagnatus granulatus (Berón de Astrada & Tomsic, 2002), Hemigrapsus oregonensis (Sztarker, Strausfeld & Tomsic, 2005), and C. maenas (Krieger et al., 2012). A clear distinction between these two neuropils is difficult because they are tightly adjoined. Therefore, a comparative volumetric analysis was impractical. Nevertheless, our preparations indicate that in Uca, the TM/HN-complex is markedly smaller in diameter compared to all other crabs being analyzed (compare Fig. 13A with Figs. 7A–7E, Figs. 9A–9C and 11A–11B). A compartmentalization of the hemiellipsoid body into one cap and 1–2 core neuropil masses is obvious in G. natalis (Fig. 9C) and in G. tiomanicum (Figs. 11B–11C2), whereas such a subdivision could not be resolved in the other crabs analyzed. According to Sandeman et al. (1992), each of these neuropils is associated with a cluster of neurons, namely cluster (4), which is located closely to the terminal medulla and cluster (5), which is adjacent to the hemiellipsoid body in decapods and contains hundreds of interneurons of minute diameter. However, in the brachyurans studied, a clear separation of these two clusters was impossible. Rather, the TM/HN -complex is surrounded by a confluent cortex of somata which therefore will be referred to as cluster (4/5) here (Figs. 7C, 9C, 11B and 13A–13B).

Median protocerebrum

The median protocerebrum is composed of the closely fused AMPN and PMPN and appears broader than long. In all brachyuran crabs studied, it has a butterfly-shape in horizontal sections (Fig. 3C). The AMPN and the PMPN are identifiable by showing distinct SYN (Figs. 7F, 7H–7J, 9D–9E, 11D–11F, 13C–13E, 15A–15D, 16A1–16C2 and 17), weaker RFA (Figs. 11D–11F, 13D–13E, 13G, 16A1–16C2 and 17) and AST in the periphery (Figs. 9D–9E, 15A–15D and 17). Although allatostaninergic and RFamidergic fibers innervate the whole brain, the V-shaped protocerebral bridge (PB) and especially the cylindrical or cigar-shaped central body (CB) further posteriorly show the densest RFA as well as AST (Figs. 7F–7I, 9E, 11D–11F, 13D, 13G, 15A–15D, 16A2–16C2 and 18) besides distinct SYN (Figs. 7H–7J, 9E, 11E–11F, 13D–13E, 15A–15D, 16A2–16C2 and 17). In sections at the level of the CB, the separation of the AMPN from the PMPN becomes visible (Figs. 7H, 9E, 11F, 11G, 13D, 13F, 15B and 17). Anterior to the PB, the nuclear marker reveals hundreds of somata of varying diameters (5–10 µm in all species studied) that are grouped within the cell cluster (6). This cluster also comprises a subset of few somata with a markedly larger diameter that display distinct RFA (diameters approx. 30 µm in C. armatum, 25 µm in G. tiomanicum, and 20 µm in U. tangeri) and AST (approx. 30 µm in E. sinuatifrons and G. natalis). The neuropils of the median protocerebrum are regularly pierced by blood vessels of the circulatory system (e.g., the cerebral artery (CA); Figs. 9D–9E 11D, 11F, 13C and 13E) and by large tracts of neurites (e.g., the projection neuron tract (PNT); Figs. 7H, 9D, 11D–11G, 13C–13E, 13G, 15B, 15D, 16A1 and 16B1–16B2) and can be inferred from the negative imprint due to the absence of immunoreactivity against the tested antisera. The projection neuron tract consists of neurites of olfactory projection neurons whose cell bodies are located in the somata clusters (10) situated posterior-lateral to each DCL. These neurites connect each DCL to the ipsilateral as well as the contralateral TM/HN-complex within the lateral protocerebrum and constitute a chiasm dorsally to the central body. The cerebral artery (CA) located between median protocerebrum (posterior to the PMPN) and deutocerebrum (anterior to the median antenna I neuropil) is identifiable by the nuclear counterstain of the perivascular cells (Figs. 9D–9E, 11F, 13F and 15A–15B) in horizontal sections. The dorsoventral course of the CA through the central brain could be confirmed in all crabs as it has been shown for C. maenas (Sandeman, 1967) but not its ramifications.

Deutocerebrum with special focus on structures of the primary olfactory pathway

Directly posterior-ventral to the PMPN and the CA, the unpaired median antenna I neuropil (MAN; Figs. 7F–7H, 9D–9E, 11D–11G, 13C–13G and 15B) is present in all brachyuran crabs studied displaying distinct SYN as well as AST and RFA. The border between PMPN and MAN is rather confluent but is identifiable due to the clear position of the CA (compare Figs. 9D and 9E).

Besides the deutocerebral chemosensory lobe (DCL) and the accessory lobe (AcN), other neuropils of the lateral deutocerebrum can be found within the confluent mass of the central brain composed of parts of the proto-, deuto- as well as the tritocerebrum (compare Fig. 17) such as the lateral antenna I neuropil (LAN; Figs. 7F, 9D, 11D, 13D–13E and 15B) and in a few preparations, the projection neuron tract neuropil (PNTN; Figs. 15B and 16A1). However, a complete, in-depth reconstruction of their definite outlines remains challenging. In all species investigated, the AcN and, in particular the DCL are the most delimited structures within the otherwise confluent brachyuran brain. The DCL is composed of several dozens (60–80 in G. sesarma, U. tangeri and C. armatum up to almost 200 in G. natalis—see Table 2 for further information) of barrel-like to conical cardridges, termed olfactory glomeruli (OG), of varying sizes (see Fig. 18). From a limited number of investigated specimens of U. tangeri, it appeared that in two males analyzed the number of olfactory glomeruli exceeded that of one female by a factor of ca. 2 (36 OG in ♂versus 76–80 OG in ♀), whereas the males featured approximately a third of the average female glomerular volume (see Table 2) resulting in an almost equal volume of the entire DCL in both sexes. In all marine brachyuran species studied, the numbers as well as the average volumes of olfactory glomeruli markedly exceed those of the co-studied terrestrial brachyurans (Table 2 and Fig. 18), though the general brain dimensions are somewhat similar (see Figs. 3 and 17). In brain sections of aquatic representatives of Brachyura (in the four exclusively marine; and to some degree, in the freshwater species E. sinuatifrons), the olfactory glomeruli are larger and more elongated compared to those of the terrestrial species studied here. A clear regionalization of each olfactory glomerulus into a cap, a subcap, and a base region (from the periphery of the DCL to its center) appears more pronounced in aquatic brachyurans than in the terrestrial species (Fig. 18). The cap and base regions show stronger SYN (Figs. 8A, 8D1, 14E, 15E, 16A1, 16B1, 17 and 18) than the subcap region in these species. In a subset of experiments, the subcap region shows distinct RFA, but RFA is weaker in the base region and is absent in the cap region (i.e., in P. gibbesi, X. hydrophilus, X. poressa, U. tangeri; Figs. 18A and 18B), whereas the subcap region shows the most distinct AST and becomes absent towards the base region in each OG (in E. sinuatifrons, Figs. 18A and 18B; G. natalis, not shown; and in C. maenas, see Krieger et al., 2012). Anteriomedial to the median foramen of each DCL, the accessory lobe (AcN) becomes visible, consisting of dozens of microglomeruli that show distinct SYN but widely lack RFA as well as AST. The diameter of the almost spherical AcN ranges from 50 µm (in U. tangeri, X. hydrophilus, X. poressa, and P. gibbesi; Figs. 14A, 14D–14E and 16A3–16C3) up to 100 µm (75 µm in C. armatum, 100 µm in G. natalis as well as G. tiomanicum; Figs. 7F, 8A, 8E, 9D, 10C, 12A, 12D and 12F). Further medial and between the PMPN and the AcN, a somata cluster of hundreds of interneurons (ca. 5–8 µm in diameter) appears. This cell cluster (9/11) is clearly revealed by the nuclear counterstaining (Figs. 8C, 9E, 10A, 10E and 15B) and contains subpopulations of several to dozens of allatostatinergic (Figs. 9E and 15B) and RF-amidergic interneurons (Figs. 7G, 12A–12C, 13D–13E, 16A1, 16A3 and 16B1) that are markedly larger in diameter (from 12 µm in U. tangeri; and 16 µm in X. hydrophilus; up to 30 µm in G. tiomanicum; i.e., see Fig. 12C). Neurites of cell cluster (9/11) enter each DCL via the median foramen (mF; Figs. 7F, 8C, 10B, 12B, 14A–14C, 15B, 16A1, 16B1 and 17). Lateroposterior to each DCL, a group of hundreds to thousands of olfactory projection neurons house their somata within cell cluster (10). These neurites of projection neurons enter the DCL via the posterior foramen (pF; Figs. 8B–8C, 9D, 10D, 12C, 14B–14C and 15B), connect with the olfactory glomeruli, exit the DCL via the median foramen (Figs. 8C, 12B, 14C, 15B, 16A1, 16B1 and 17), and finally project to the ipsilateral as well as the contralateral TM/HN-complex by forming a chiasm at the dorsal level of the central body (not shown). The entirety of neurites of projection neurons constitute the projection neuron tract (PNT) whose somata are housed within cell cluster (10). According to its position, the projection neuron tract neuropil (PNTN) becomes visible medial to the mF in a few preparations showing distinct SYN (Figs. 15B, 16A1 and 17).

Figure 8 Optical horizontal sections and 3D-reconstruction of deutrocerebral chemosensory lobe (DCL) in Cardisoma armatum.

(A–C) Inverted single-channel micrographs of DCL. White arrows in B mark axons of projection neurons. (D) and (D1): Detailed picture of the olfactory glomeruli (OG) with double-labeling in D and inverted single-channel picture in D1. (E) 3D-reconstruction of DCL, olfactory glomeruli and accessory neuropil (AcN) shown in four different orientations. 1: from dorsal. 2: from anterior. Dashed line represents the horizon of section given in A. 3: from posterior. Dashed line indicates the posterior foramen (pF). 4: centro-lateral view. Dashed line indicates the median foramen (mF). Abbreviations of immunhistochemical labelings and histochemical marker: NUC, nuclear marker (black); PHA, actin-labeling using phalloidin (black); RFA, labeling against RFamide (green); SYN, labeling against synapsin (magenta or black). Other abbreviation: 10 and 9/11, cell clusters (10) and (9/11); a, anterior; Base, base domain of OG; Cap, cap domain of OG; d, dorsal; l, lateral; m, median; Subcap, subcap domain of OG.

Figure 9 Micrographs of triple-labeled vibratome sections of central brain and lateral protocerebrum in Gecarcoidea natalis.

(A–C) Visual neuropils and lateral protocerebrum. Note that in A and C, two out of three channels are shown while B shows an inverted single-channel micrograph. (D and E) show two triple-labeled horizontal vibratome sections of central brain (D) and further ventral of central brain (E). Dashed line in E indicates the cerebral artery. Abbreviations of immunhistochemical labelings and histochemical marker: AST, labeling against allatostatin (green); NUC, nuclear marker (cyan); RFA, labeling against RFamide (black); SYN, labeling against synapsin (magenta). Other abbreviations: 1, 2, 3, 4/5, 6, 9/11, and 10, cell clusters (1), (2), (3), (4/5), (6), (9/11), and (10); AINv, antenna I nerve; AIINv, antenna II nerve; AcN, accessory neuropil; AMPN, anterior medial protocerebral neuropil; AnN, antenna II neuropil; CA, cerebral artery; CB, central body; DCL, deutocerebral chemosensory lobe; iCh, inner visual chiasm; HN, hemiellipsoid body; La, lamina; LAN, lateral antenna I neuropil; Lo, lobula; MAN, median antenna I neuropil; Me, medulla; oCh, outer visual chiasm; PMPN, posterior medial protocerebral neuropil; PNT, projection neuron tract; PT, protocerebral tract; TM, terminal medulla; VT, visual tract. Scale bars = 250 µm.

Figure 10 Vibratomy of double-labeled horizontal sections the deutocerebral chemosensory lobe (DCL) in Gecarcoidea natalis.

(A–D) DCL featuring olfactory glomeruli (OG). Note that only an inverted single-channel micrograph is given in (E) showing nuclear staining in the periphery of DCL. Abbreviations of immunhistochemical labeling and histochemical staining: NUC, nuclear marker (cyan or black); SYN, labeling against synapsin (magenta). Other abbreviations: 10 and 9/11, cell clusters (10) and (9/11); AcN, accessory neuropil; mF, median foramen; OG, olfactory glomerulus; pF, posterior foramen. Scale bars = 100 µm.

Figure 11 Triple-labeled micrographs of optical horizontal sections showing the central brain, lateral protocerebrum, and specific brain areas in Geosesarma tiomanicum.

(A–C) Micrographs of optical sections of visual neuropils (A) and the TM/HN-complex (B, C1 and C2). Note that in C1 and C2, inverted single-channel micrographs are shown. (D–E) Brain (D) and central body and adjacent protocerebral and deutocerebral neuropils are given in (E) and (F)in higher detail. Arrows with dashed lines in D mark giant neurons featuring distinct RFA-like immunoreactivity. Abbreviations of immunhistochemical labelings and histochemical markers: NUC, nuclear marker (cyan); PHA, actin-marker using phalloidin (green); RFA, labeling against RFamide (green or black); SYN, labeling against synapsin (magenta or black). Other abbreviations: 2, 3, and 4/5, cell clusters (2), (3), and (4/5); AINv, antenna I nerve; AMPN, anterior medial protocerebral neuropil; AnN, antenna II neuropil; CA, cerebral artery; Cap, cap neuropil; CB, central body; DCL, deutocerebral chemosensory lobe; HN, hemiellipsoid body; Lal, lateral accessory lobe; LAN, lateral antenna I neuropil; Lo, lobula; MAN, median antenna I neuropil; Me, medulla; PB, protocerebral bridge; PMPN, posterior medial protocerebral neuropil; PNT, projection neuron tract; PT, protocerebral tract; TM, terminal medulla; VT, visual tract.

Figure 12 Optical horizontal sections and 3D-reconstruction of deutrocerebral chemosensory lobe (DCL) in Geosesarma tiomanicum.

(A–E) Triple labeled optical sections of DCL and adjacent neuropils and cell clusters. Deutocerebral accessory lobe (AcN) is shown in D. Olfactory glomeruli (OG) are indicated by white dashed lines in E in higher detail. (F) 3D-reconstruction of DCL, olfactory glomeruli, and accessory neuropil in four different orientations—1: from dorsal. 2: centro-lateral view. White dashed line indicates orientation of section given in A. Black dashed line indicates median foramen (mF). 3: from ventral. 4: from posterior. Dashed line highlights posterior foramen (pF). Abbreviations of the immunhistochemical labelings and histochemical marker: NUC, nuclear marker (cyan); RFA, labeling against RFamide (green); SYN, labeling against synapsin (magenta). Other abbreviations: 10, 9/11, and 14/15, cell cluster (10), (9/11), and (14/15); a, anterior; AMPN, anterior medial protocerebral neuropil; d, dorsal; DCLa, anterior sublobe of the DCL; DCLl, lateral sublobe of the DCL; l, lateral; m, median. Scale bars = 100 µm.

Figure 13 Triple-labeled optical horizontal sections of central brain, lateral protocerebrum, and specific brain areas in Uca tangeri.

(A and B) Vertical section showing the visual neuropils (A) and the TM/HN-complex (B). (C–G) show vertical sections of the brain and specific brain areas. Note that in C, F, and G, inverted single-channel micrographs are displayed. Arrows with dashed lines in D and G mark large neurons in cell cluster (6) featuring distinct RFA-like immunoreactivity.Abbreviations of immunhistochemical labelings and histochemical markers: NUC, nuclear marker (cyan or black); RFA, labeling against RFamide (green or black); SYN, labeling against synapsin (magenta or black). Other abbreviations: 2, 3, 5, 6, 10, and 9/11, cell clusters (2), (3), (5), (10), and (9/11); AMPN, anterior medial protocerebral neuropil; AnN, antenna II neuropil; CA, cerebral artery; CB, central body; DCL, deutocerebral chemosensory lobe; HN, hemiellipsoid body; iCh, inner visual chiasm; Lal, lateral accessory neuropil; LAN, lateral antenna I neuropil; Lo, lobula; MAN, median antenna I neuropil; Me, medulla; PB, protocerebral bridge; PMPN, posterior medial protocerebral neuropil; PNT, projection neuron tract; PT, protocerebral tract; TM, terminal medulla; VT, visual tract. Scale bars, 250 µm.

Figure 14 Optical horizontal sections and 3D-reconstruction of deutocerebral chemosensory lobe (DCL) in Uca tangeri.

(A, C and E) Inverted single-channel micrographs of DCL. Dashed lines in E indicate two olfactory glomeruli (OG). (B and D) Triple-labeled optical sections of DCL in B and accessory lobe (AcN) in D. (F) 3D-reconstruction of DCL, its OG and AcN in four different perspectives. 1: from dorsal. 2: from anterior. Dashed line represents horizon of section represented in A. 3: from posterior. Dashed line outlines posterior foramen (pF). 4: centro-median view. Dashed line highlights median foramen (mF). Abbreviations of immunhistochemical labelings and histochemical markers: NUC, nuclear marker (cyan); RFA, labeling against RFamide (green or black); SYN, labeling against synapsin (magenta or black). Abbreviation: 10, cell cluster (10); a, anterior; d, dorsal; l, lateral; m, median; mF, median foramen; PNT, projection neuron tract.

Figure 15 Vibratomy of triple-labeled horizontal vibratome sections of central brain and specific brain areas in Epilobocera sinuatifrons.

(A and B) Two brain sections (100 µm) from dorsal (A) to ventral (B) are shown. Arrows with dashed lines in A and B point at specific neurons within cell cluster (6) featuring distinct AST-like immunoreactivity. Dashed line highlights the position of the projection neuron tract (PNT). (C and D) show neuropils of the central complex from dorsal (C) to ventral (D) in more detail. Higher detailed insight from deutocerebral chemosensory lobe (DCL) is given in (E). Dashed line outlines a single olfactory glomerulus (OG). Abbreviations of immunhistochemical labelings and histochemical markers: AST, labeling against allatostatin (green); NUC, nuclear marker (cyan); SYN, labeling against synapsin (magenta). Other abbreviations: 6, 10, and 9/11, cell clusters (6), (10), and (9/11); AMPN, anterior medial protocerebral neuropil; AnN, antenna II neuropil; Base, base domain of OG; CB, central body; Cap, cap domain of OG; DC, deutocerebrum; Lal, lateral accessory neuropil; LAN, lateral antenna I neuropil; MAN, median antenna I neuropil; mF, median foramen; PB, protocerebral bridge; pF, posterior foramen; PMPN, posterior medial protocerebral neuropil; PNTN, projection neuron tract neuropil; Subcap, subcap domain of OG.

Figure 16 Vibratomy of triple-labeled horizontal sections (100 µ m) of central brains and specific brain areas in Xantho hydrophilus, Xantho poressa, and Perrcnon gibbesi.

Note that species are represented column by column (A–C). Comparable brain areas are given line by line (1–3). (A1–C1) display equally scaled micrographs of horizontal vibratome sections of one hemisphere per species. (A2a–C2) show neuropils of central complex in more detail. Arrow with a dashed line in A2a identifies one of a subset of somata within cell cluster (6) featuring distinct RFA-like immunoreactivity. (A3–C3) display neuropils and somata of primary olfactory pathway in deutocerebrum (DC). Abbreviations of immunhistochemical labelings and histochemical markers: NUC, nuclear marker (cyan); RFA, labelling against RFamide (green); SYN, labelling against synapsin (magenta). Other abbreviations: 6, 10, 9/11, and 14/15, cell clusters (6), (10), (9/11), and (14/15); AcN, accessory neuropil; AMPN, anterior medial protocerebral neuropil; CA, cerebral artery; CB, central body; DCL, deutocerebral chemosensory lobe; lF, lateral foramen; mF, median foramen; mPC, median protocerebrum; OG, olfactory glomerulus; PB, protocerebral bridge; PMPN, posterior medial protocerebral neuropil; PNT, projection neuron tract; PNTN, projection neuron tract neuropil; TC, tritocerebrum. Scale bars = 100 µm.

Figure 17 Collage of triple-labeled and equally scaled optical sections of brain hemispheres in Carcinus maenas, Percnon gibbesi, Xantho hydrophilus, Xantho poressa, Epilobocera sinuatifrons, Cardisoma armatum, Gecarcoidea natalis, Geosesarma tiomanicum, and Uca tangeri.

The schematic drawing of the brain hemisphere in C. maenas (dorsal view) is modified from Krieger et al. (2012). Abbreviations of immunhistchemical labelings and histochemical markers: NUC, nuclear marker (cyan); RFA, labeling against RFamide (green or black); SYN, labeling against synapsin (magenta or black); AST, labeling against allatostatin (green). 9/11 and 10, cell clusters (9/11) and (10); Other abbreviations: CB, central body; DC, deutocerebrum; DCL, deutocerebral chemosensory lobe; mPC, median protocerebrum; PNTN, projection neuron tract neuropil; TC, tritocerebrum. Note that for the comparison of sizes, the pairings of brain hemispheres are summarized somewhat arbitrarily, showing the exclusive marine species and the freshwater crab E. sinuatifrons on the upper two panels, and the four land crabs are displayed below.

Figure 18 Collage of double-labeled and equally scaled horizontal sections of deutocerebral chemosensory lobe (DCL in A) and its olfactory glomeruli (OG in B) in all species studied.

Deutocerebral chemosensory lobes and their olfactory glomeruli of exclusively marine species are shown followed by the neuropils of the freshwater brachyuran Epilobocera sinuatifrons and those of brachyuran species featuring different degrees of terrestrialization (in A and B, respectively). Abbreviations of immunhistochemical labelings and histochemical markers: PHA, actin labeling using phalloidin; RFA, labeling against RFamide (green); SYN, labeling against synapsin (magenta).

Tritocerebrum

The tritocerebral antenna II neuropil (AnN) and further dorsally the tegumentary neuropil (TN) compose the posteriormost parts of the central brain, being located anterolaterally to the esophagus. An identification of the neuropil borders is difficult due to their confluent connection to the deutocerebrum. The AnN that receives chemosensory as well as mechanosensory input from the second antenna is identifiable in a few preparations by tracing back the course of the antenna II nerve (AIINv; Figs. 9D and 9E). Since we were unable to trace back the course of the presumably thin tegumentary nerve (TNv), the precise position and shape of the tegumentary neuropil remains uncertain.

Discussion

In this study, we compare the neuroanatomy of the brain in four brachyurans that display different levels of terrestrial adaptations using the antisera against presynaptic proteins, the neuropeptides FMRFamide, and allatostatin as well as markers for actin and DNA. In the following, we will compare and discuss the results of these four brachyuran species with each other as well as with one freshwater and four marine brachyurans. Special attention is given to the primary olfactory system and related structures to highlight differences between terrestrial brachyurans and their aquatic relatives.

In contrast to other reptant Malacostraca such as Anomala, which display a clear separation of their deutocerebral neuropils (e.g., Harzsch & Hansson, 2008; Krieger et al., 2010; Krieger et al., 2012), these neuropils are widely confluent and therefore often become indistinguishable in brachyurans. Sandeman, Scholtz & Sandeman (1993) and Krieger et al. (2012) discussed the possible connection between brain “condensation”, the fusion of synaptic neuropils, and evolutionary success in these groups. The condensation of nervous tissue may have coincided with a process that is sometimes called “carcinisation” (Borradaile, 1916), or “brachyurisation” (Števčić, 1971). These synonyms circumscribe a hypothesis of how the condensed crab shape may have developed (McLaughlin & Lemaitre, 1997), both concerning the overall brachyuran habitus as well as internal consolidation of organs like the fusion of the first three ganglia of the ventral nerve cord into one joint complex (Števčić, 1971). According to Števčić (1971), it was also assumed that this process mainly leads to a more complex behavior and better coordination in semiterrestrial and terrestrial crabs, since neuropil condensation and shortening of connections within the central nervous system may improve the performance of the system, e.g., in terms of processing speed. The fusion is most conspicuous in the posterior part of the brain, where the neuropils of the deutocerebrum adjoin those of the tritocerebrum.

Visual ecology and the protocerebrum

Terrestrial brachyurans have been prime examples to study visual ecology in crustaceans (reviews by Zeil & Hemmi, 2006; Zeil & Hemmi, 2014; Hemmi & Tomsic, 2012). Visual orientation has been very well studied in members of the genus Uca but poorly in any of the other ocypodid species (Zeil & Hemmi, 2014 and references therein). Field experiments for individuals of U. tangeri have shown that they can recognize predators at greater distance, triggering an escape behavior. In addition, the animals react to their own mirror image and can visually distinguish the gender of their conspecifics (Altevogt, 1957; Altevogt, 1959; Von Hagen, 1962; Korte, 1965; Land & Layne, 1995; Zeil & Al-Mutairi, 1996). Representatives of the genus Uca can also distinguish colors (Korte, 1965; Hyatt, 1975; Detto, 2007), which is an important factor for social interactions (Detto et al., 2006; Detto, 2007). Ultraviolet light, for example, is reflected by the claw of Uca-males which attracts females (Detto & Backwell, 2009). Aspects of homing and path integration were also thoroughly analyzed in members of the genus Uca (e.g., Hemmi & Zeil, 2003; Layne, Barnes & Duncan, 2003a; Layne, Barnes & Duncan, 2003b; Walls & Layne, 2009). Clearly, vision plays an essential role in the ecology of Uca.

In all species examined here, the neuropils of the lateral protocerebrum are located within the eyestalks in some distance to the central brain (compare Sandeman et al., 1992; Sandeman, Scholtz & Sandeman, 1993). In this study, the three visual neuropils (lamina, medulla, and lobula) could be identified in all individuals of the different species, and their location and anatomy matches that of other described brachyuran species (Tsvileneva, Titova & Kvashina, 1985; Sandeman et al., 1992; Sandeman, Scholtz & Sandeman, 1993; Sztarker, Strausfeld & Tomsic, 2005; Sztarker et al., 2009; Krieger et al., 2012; Berón de Astrada et al., 2013). However, the small lobula plate, the fourth visual neuropil, could not be found in any of the analyzed species, most likely because of technical difficulties but was previously identified in other brachyuran species such as Chasmagnathus granulatus, Hemigrapsus oregonensis (Sztarker, Strausfeld & Tomsic, 2005; Sztarker et al., 2009) and C. maenas (Krieger et al., 2012). The terminal medulla (or medulla terminalis) and the hemiellipsoid body are considered to function as secondary higher-order neuropils (Wolff et al., 2012; Wolff & Strausfeld, 2015). They integrate different modalities such as visual and olfactory information that were already preprocessed in the primary sensory brain centers (visual neuropils and deutocerebral chemosensory lobes; reviewed in Schmidt, in press). Furthermore, the TM/HN-complex receives input from the ventral nerve cord (VNC) and other regions of the central brain. The terminal medulla and especially the hemiellipsoid body are also referred to as centers of learning and memory that functionally correspond to the mushroom bodies in hexapods. For the brachyurans studied here, there were only few species-specific differences visible in our preparations. We conclude that all terrestrial brachyurans examined here have well developed visual neuropils. Thus, they possess a neuronal substrate for a sophisticated analysis of the compound eye input. Therefore, as in their marine counterparts, visual cues most likely play important roles in the terrestrial brachyurans’ behaviors such as food search, mating, and orientation.

Chemical senses: the peripheral olfactory pathway

It is well established that marine crustaceans use chemical cues to locate mates, signal dominance, recognize individual conspecifics, find favored foods and appropriate habitats, and assess threats such as the presence of predators (reviews e.g., Derby et al., 2001; Grasso & Basil, 2002; Derby & Sorensen, 2008; Thiel & Breithaupt, 2011; Wyatt, 2011; Derby & Weissburg, 2014). Malacostracan crustaceans that live in aquatic habitats use several systems for detecting chemicals, and these are distributed over their body surface, walking appendages, and mouthparts. We will focus our discussion on those sensilla concentrated on the two pairs of antennae (reviews e.g., Hallberg, Johansson & Elofsson, 1992; Hallberg & Skog, 2011; Schmidt & Mellon, 2011). The first antennal pair (the antennules) is equipped with specialized olfactory sensilla (aesthetascs) in addition to bimodal chemo- and mechanosensilla, functioning as contact-chemoreceptors, whereas the second pair of antennae is only equipped with the latter. The tips of the first antennae (more specifically the lateral flagellum) bear a tuft region with arrays of aesthetascs that house branched dendrites of olfactory sensory neurons (reviews by Hallberg, Johansson & Elofsson, 1992; Hallberg & Hansson, 1999; Mellon Jr, 2007; Hallberg & Skog, 2011; Schmidt & Mellon, 2011; Derby & Weissburg, 2014). There are multiple studies on the ultrastructure of these aesthetascs (e.g., Ghiradella, Case & Cronshaw, 1968a; Ghiradella, Case & Cronshaw, 1968b; Snow, 1973; Wasserthal & Seibt, 1976; Tierney, Thompson & Dunham, 1986; Spencer & Linberg, 1986; Grünert & Ache, 1988; Gleeson, McDowell & Aldrich, 1996), but unfortunately none of these studies includes any of the species analyzed in the present work.

We observed that all four terrestrial brachyurans studied here have shorter antennae in relation to their body sizes, and feature markedly fewer and shorter aesthetascs compared to their marine relatives (Table 2; Figs. 3A, 3B and 4–5). These findings suggest that possessing short and hidden first antennae equipped with few, short, and blunt aesthetascs seems to be a shared feature and most likely a specific adaptation in all terrestrial brachyurans. We suggest that this feature may be an adaptation to minimize water loss across the cuticle. Furthermore, a typical marine (brachyuran) array of long and slender aesthetascs will likely collapse out of water and most likely will be non-functional on land. Studies on other terrestrial crustacean taxa such as representatives of the Isopoda and Anomala support the idea that all terrestrial crustaceans share a size reduction of antennal sensilla including the aesthetascs (compare Hansson et al., 2011). The aesthetascs of terrestrial hermit crabs of the taxon Coenobitidae, for example, display striking differences to those of marine hermit crabs in that they appear short and blunt (compare Ghiradella, Case & Cronshaw, 1968a; Stensmyr et al., 2005). In robber crabs, Birgus latro, the largest known land arthropods, they are confined to the ventral side of the primary flagella and are flanked by presumably bimodal contact-chemoreceptive sensilla. A preliminary analysis using classical histology and transmission electron microscopy (TEM) revealed that, in contrast to marine crustaceans, the aesthetascs of Coenobitidae have an asymmetric profile, with the protected side lined with a thick cuticle (Tuchina et al., 2015). The exposed side is covered with a thinner cuticle, a feature that most likely is necessary to enable the passage of odors (Stensmyr et al., 2005). These and other morphological features were interpreted as mechanisms to minimize water evaporation while maintaining the ability to detect volatile odorants in gaseous phase (Stensmyr et al., 2005). Furthermore, antennal olfaction at least in coenobitids is assumed to depend on activity of the asthetasc-associated epidermal glands discharging their secretion to the base of related aesthetascs. By the aid of the mucous secretion covering the entire thinner cuticle, aesthetascs are provided with a moist, sticky layer essential for binding, sampling, and finally perceiving (after transcuticular passage) volatile odors (Tuchina et al., 2014).

However, in terrestrial Anomala, contrary to terrestrial Brachyura, the first antennae are extensively enlarged and the number of aesthetascs markedly increased as compared to marine representatives (Table 2), and there is evidence that Coenobitidae may have evolved good terrestrial olfactory abilities (Greenaway, 2003). In fact, behavioral studies have suggested that these animals are very effective in detecting food from a distance and in responding to volatile odors (Rittschof & Sutherland, 1986; Vannini & Ferretti, 1997; Stensmyr et al., 2005). These omnivorous crabs are attracted by volatiles emitted by many different sources such as seawater, wellwater, distilled water (Vannini & Ferretti, 1997), crushed conspecifics or snails (Thacker, 1994), fruits, seeds, flowers (Rittschof & Sutherland, 1986; Thacker, 1996; Thacker, 1998), and finally even horse faeces and human urine (Rittschof & Sutherland, 1986). By conducting a two-choice bioassay with Coenobita clypeatus using an arena with a centrally placed shelter with two pit-falls on each side, Krång et al. (2012) found that the animals were strongly attracted to natural odors from banana and apple. Furthermore, wind-tunnel experiments with C. clypeatus suggest that these animals display a behavior that may be described as odor-gated anemotaxis (C Mißbach, J Krieger, S Harzsch, BS Hansson, 2015, unpublished results). Furthermore, electrophysiological studies using electroantennograms in B. latro confirmed that the aesthetascs respond to volatile substances (Stensmyr et al., 2005). In aquatic crustaceans, antennular flicking enhances odorant capture by shedding the boundary layer (Koehl, 2011; Reidenbach & Koehl, 2011; Mellon Jr & Reidenbach, 2012). Coenobitidae also show flicking behavior similar to that seen in their marine relatives, thus maximizing odor sampling (Stensmyr et al., 2005). Mellon Jr & Reidenbach (2012) suggested that considering the higher kinematic viscosity of air versus water and the resulting lower Reynolds numbers, the aesthetascs of Coenobitidae nevertheless operate in a range where boundary layer shedding could be effectively achieved by antennular flicking. Taken together, these behavioral and morphological observations suggest that terrestrial Anomala evolved aerial olfaction and actively use their first pair of antennae to detect volatile odors.

In contrast to this highly sophisticated olfaction-related behavior of Coenobitidae, our limited observations in the laboratory of the terrestrial brachyurans C. armatum, G. tiomanicum, and U. tangeri suggest that their first pair of antennae extended and that flicking behavior occurred only if animals were immersed in water but the antennae were not exposed to the air. This holds also true for the second pair of antennae, except for Uca tangeri. In Gecarcoidea natalis, we did not observe that the first as well as the second pair of antennae were exposed in their terrestrial habitat as observed in three field trips to Christmas Island (J Krieger, MM Drew, S Harzsch, BS Hansson, 2012, unpublished obs.). These animals enter the water only during the spawning season (Orchard, 2012). As laid out above, crabs orient very well on land, and many studies have suggested vision to be the dominating sense in terrestrial Brachyura. Our morphological results and preliminary behavioral observations suggest that, contrary to Anomala, the detection of volatile substances plays only a minor role in the sensory ecology of Brachyura while on land. If it holds true that the first antennae in brachyurans are only functional in an aquatic environment, we may expect to see this reflected in the organization of primary processing areas within the brain. With respect to the critical cost-benefit ratio of maintaining the highly energy-demanding nervous tissue, providing processing capacities for poorly used sensory modalities may be too costly, so that these brain areas become reduced during evolution.

Chemical senses: the central olfactory pathway

The chemosensory neurons associated with the aesthetascs versus the bimodal non-aesthetasc sensillae (contact-chemoreceptors) on the first antennae of malacostracan crustaceans innervate distinct regions in the brain (see review of Schmidt & Mellon, 2011; Derby & Weissburg, 2014). The axons of the olfactory sensory neurons (OSNs) associated with the aesthetascs target the deutocerebral chemosensory lobes (in our previous studies termed olfactory lobes), whereas the axons associated with non-aesthetasc sensilla innervate the lateral antenna 1 neuropil (LAN; for other crustacean chemosensory systems see Schmidt & Mellon, 2011). For all species studied in this paper, the deutocerebral chemosensory lobes of the deutocerebrum, the accessory neuropils, the lateral antenna I neuropil and the median antenna I neuropil were well identifiable. Their structure and arrangement corresponds to that described in other Brachyura (Sandeman et al., 1992; Krieger et al., 2012). Also, the projection neuron tract and the cerebral artery could be depicted as characteristic landmarks. In all species investigated, the deutocerebral chemosensory lobes (DCL) share the typical malacostracan organization, featuring a radial array of barrel- to wedge-shaped olfactory glomeruli that form the thick synaptic layer of the lobe, with their apices pointing inwards (compare Schachtner, Schmidt & Homberg, 2005; Schmidt & Mellon, 2011). Although the DCLs of the species studied here have a similar overall organization, the relative size of the DCL to the central brain displays the most striking difference between aquatic and terrestrial brachyurans. While in all aquatic brachyurans studied, the DCLs are comparably large, they are conspicuously much smaller within terrestrial brachyurans, an observation also made in the land crab Chiromantes haematocheir (Honma et al., 1996). This relation also applies to the number and size (length especially) of olfactory glomeruli which are higher in all aquatic brachyuran species studied (see Table 2). These morphological aspects seem to be strongly correlated with the reduction of aesthetasc number and size as discussed above and therefore may represent another adaptation to terrestrialization (compare Figs. 17 and 18). However, a linear correlation between the number of aesthetascs and number of olfactory glomeruli could not be identified here, which is in accordance with the varying convergence ratios (aesthetascs/glomeruli) reported by Beltz et al. (2003). In summary, morphometric quantifications of neuronal structures have indeed to be considered as rough estimates to infer sensory processing performance of a species, and the species-dependent lifestyles play of course a large role for the evaluation of olfactory capacity.

Although we have analyzed admittedly only few specimens (two males and one female), our findings nevertheless hint at a sexual dimorphism of the DCL of U. tangeri. In addition to numerous reports of sexual dimorphism of insect brains (e.g., Koontz & Schneider, 1987; Homberg, Christensen & Hildebrand, 1989; Rospars & Hildebrand, 2000; Jundi et al., 2009; Streinzer et al., 2013; Montgomery & Ott, 2015) especially of the primary olfactory system, such a sexual dimorphism in crustaceans is well described from the DCLs in Euphausiacea and Mysidacea (Johansson & Hallberg, 1992). Furthermore, Loesel (2004) suggested a sexual dimorphism in central body architecture in the genus Uca. However, further investigation of sexual dimorphic features within the brain of crustaceans is crucial to understand the general principles in crustacean communication and their underlying structures. Their pronounced sex-specific external morphology regarding courtship behavior (e.g., the conspiciuous heterochely and eye stalk extensions in males) indicates that representatives of the genus Uca can serve as favorable study organisms to explore such aspects.

It has been well documented in aquatic malacostracans including crayfish, clawed and clawless lobsters, marine brachyurans, and hermit crabs (Schachtner, Schmidt & Homberg, 2005; Schmidt & Mellon, 2011; Krieger et al., 2012; Polanska et al., 2012) that the olfactory glomeruli are regionalized along their long axis to provide an outer cap, a subcap, and a base region. The subcap region of decapod olfactory glomeruli displays another level of subdivision when viewed in cross-sections and is separated into a central rod, a core region, and an outer ring. These patterns of subdivision of decapod olfactory glomeruli have been suggested to mirror a functional subdivision (Schmidt & Ache, 1997). Such a regionalization was not very obvious in the terrestrial brachyuran glomeruli which we analyzed. In conclusion, it seems obvious that the reduced sensory input to the deutocerebral sensory lobe in terrestrial brachyurans decreases the processing demands in the system, which in turn is reflected in the small size of olfactory glomeruli in addition to the lowered structural and functional complexity therein. These findings are also supported by the behavioral observations described above and support the idea that, while on land, olfaction is subordinate to vision in brachyurans. Along these lines, neuroanatomical studies of the olfactory system in marine versus terrestrial isopod crustaceans also suggested that in the terrestrial animals the deutocerebral chemosensory system has lost some of its importance during the evolutionary transition from water to land (Harzsch et al., 2011; Kenning & Harzsch, 2013).

Contrarywise, neuroanatomical studies analyzing the central olfactory pathway in terrestrial Anomala including Coenobita clypeatus (Harzsch & Hansson, 2008; Polanska et al., 2012; Wolff et al., 2012), and Birgus latro (Krieger et al., 2010) in comparison to several marine anomalan taxa of the subgroup Paguroidea (Krieger et al., 2012) suggested that in both terrestrial species, the primary olfactory centers targeted by antenna 1 aesthetasc afferents strongly dominate the brain and display conspicuous side lobes that are not present in the marine representatives, suggesting that a significant elaboration of brain areas involved in olfactory processing has taken place. The DCLs are markedly enlarged, and the number of olfactory glomeruli is increased compared to other marine anomalans studied (Table 2).

The tritocerebrum: antenna II neuropil and flow detection

In arthropods, the detection of flow is essential for tracking odor sources but also for anemotaxis, and in crustaceans, antenna 2 most likely plays a major role in detecting flow. In many malacostracan crustaceans, the second pair of antennae bear mostly mechanosensory sensilla as well as bimodal chemo- and mechanosensory sensilla (Schmidt & Mellon, 2011) presumably working as contact-chemoreceptors. This pair of appendages is associated with the tritocerebral neuromere, and its afferents target the bilaterally paired antenna 2 neuropils (AnN) that extend posterolaterally to either side of the esophageal foramen (Figs. 9D–9E, 11D, 13E and 15B). In some representatives of Decapoda, this neuropil is transversely divided into segment-like synaptic fields, suggesting a somato- or spatiotopic representation of the mechanoreceptors along the length of the second antenna (reviewed in Krieger et al., 2012). In the marine anomalan Pagurus bernhardus, this neuropil is elongate and the transverse segmentation is very obvious (Krieger et al., 2012), enforcing the idea that the sensory array of antenna 2 may be mapped along its length. In terrestrial Anomala, the antenna 2 neuropils of C. clypeatus and B. latro are rather inconspicuous, as is a transverse segmentation (Harzsch & Hansson, 2008; Krieger et al., 2010). In isopods, which are considered the most successful terrestrialized crustaceans, a transverse segmentation of the prominent AnN, as has been shown for hermit crabs, is clearly identifiable in marine but indistinct in terrestrial isopods (Harzsch et al., 2011). Since in terrestrial isopods, the first pair of antennae is highly reduced in size and the associated DCL seems to be absent, the idea arose that the pronounced second pair of antennae and its associated antenna 2 neuropils together may function as the major sensory organ (Harzsch et al., 2011; Kenning & Harzsch, 2013). Contrarywise, in both marine (Krieger et al., 2012) as well as terrestrial brachyurans, the antenna 2 neuropil is part of a large neuropil mass that is composed of both deuto- and tritocerebral portions, thus making the antenna 2 neuropil hardly identifiable. A transverse segmentation has not been detected so far, neither in marine nor in terrestrial brachyurans. From these data and especially from behavioral observations in Brachyura, one could argue that flow detection on land would have to be realized by other body parts rather than in the second pair of antennae. Undoubtedly, further analyses are required to clarify the functional relevance of the second pair of antennae and to check for structural as well as functional differences that may represent adaptations for detecting flow in water versus in air.

Conclusions

During a relatively short evolutionary time period, several crustacean lineages have convergently adapted to a number of highly diverse terrestrial habitats in which they have become highly successful (reviews of Bliss & Mantel, 1968; Powers & Bliss, 1983; Greenaway, 1988; Greenaway, 1999; Hartnoll, 1988). We are interested in which crustacean lineages successfully evolved aerial olfaction during this evolutionary transition (Hansson et al., 2011). As far as isopod crustaceans are concerned, it appears that their deutocerebral neuronal substrate for distance olfaction has largely eroded away in the terrestrial species, whereas there is good evidence for contact chemoreception using the tritocerebral pair of antennae (e.g., Harzsch et al., 2011; Kenning & Harzsch, 2013). For representatives of the Coenobitidae (Anomala), however, there is compelling evidence from neuranatomical, physiological, transcriptomic, and behavioral studies that aerial olfaction plays a major role in the animal’s behavioral repertoire (Harzsch & Hansson, 2008; Krång et al., 2012; Polanska et al., 2012; Groh et al., 2014; Tuchina et al., 2014). Brachyura take an intermediate position, and the question arises which aspects of the terrestrial olfactory landscape they are able to detect with their reduced peripheral and central olfactory pathway. In addition to volatile chemicals, humidity and CO2-concentration may be crucial cues for these animals. As with terrestrial species analyzed, a clear correlation between the specific brain anatomy and the degree of terrestriality could not be deduced. Furthermore, our study raises the possibility that in the semi-aquatic Uca tangeri sex-specific differences regarding average size and number of olfactory glomeruli exist, which may indicate that communication via sexual pheromones could be a possible function in some land crab species. Alternatively, those taxa with a mostly amphibious lifestyle may use their olfactory system while submersed in water. This idea most likely does not apply to G. natalis which, when immersed in water for a short amount of time will drown. Since there are more brachyuran taxa that independently succeeded in evolving a terrestrial life-style than those representatives examined here (Fig. 1), an ongoing comparative analysis of brachyuran neuroanatomy remains an exciting topic. Therefore, further studies to evaluate and compare general aspects of terrestrialization within brachyurans as well as with those of other crustacean lineages are promising.

Only a few lineages within Crustacea have independently evolved terrestrial olfaction to different degrees, suggesting that the evolution of effective olfactory systems (or sensory systems more generally) on land is highly challenging. As for amphibious olfaction in secondarily aquatic insects, Hodgson (1953) reported that in the amphibious beetle, Laccophilus maculosus, specimens are capable to perceive the same chemical compounds in air as well as under water, even though sensitivity in air is increased by a factor of 5–10 in comparison to underwater sensitivity. Supported by morphological data, bioassays using antennal ablations in L. maculosus suggest an amphibious chemoreception which is most likely based on the antennal sensilla basiconica (Hodgson, 1953). However, an effective amphibian olfaction in brachyurans with terrestrial adaptations like in the secondarily aquatic L. maculosus could not be verified here.

When comparing different taxa within Crustacea that conquered land, it becomes obvious that they feature a variety of terrestrial adaptations to different degrees; and from a scientific point of view, the objective to evaluate those different degrees is logical. The proposed levels of terrestrial adaptation from T1 to T5 for land crabs after Powers & Bliss, (1983) are commonly used to date, but it was also reported that this classification is “far from perfect” (see review Hartnoll, 1988). Derived from general biological considerations, this classification features five gradual levels of terrestrialness. These features include the time of day and the total periods spent actively on land (intertidal species), the requirement of regular immersion or drinking of water, and the (sea) water-dependency for larval development. Although this classification comprised several aspects of terrestriality, it is not possible to assign each species to a unique level in either case because the conquest of land is a gradual process that demands for diverse different adaptations. Therefore, Schubert and co-workers (2000) proposed three simplified degrees of terrestrialization referring to adult life in addition to larval development as follows: (A) terrestrial adults with marine larvae, (B) limnic adults with marine larvae, and (C) adults that breed in inland waters and hence are independent from the ocean (e.g., several Sesamidae). If we take a perspective solely related to deutocerebral olfaction (as mediated by the first pair of antennae), the levels of terrestrial adaptation may be grouped as TO0—not functional at all (terrestrial isopods); TO1—functional in water and non-functional on land (as suggested for C. armatum); TO2—functional in water as well as on land => amphibious (needs to be tested like in the amphibious beetle L. maculosus); and TO3 not functional in water but functional on land (Coenobitidae and presumably G. natalis). Although it is beyond the scope of this paper to propose an alternative extensive classification system, we nevertheless conclude that the existing classification systems must be improved to describe the degree of terrestrial adaptation for an animal as a whole. In fact, it seems crucial that multiple biological aspects such as development, mating, foraging, biorhythm, physiology as well as anatomy be taken into account for an adequate evaluation of terrestriality of each species.

We would like to express our gratitude to Guido Dehnhardt and the staff of the Marine Science Center in Rostock for free provision of diving equipment and the permission to sample Carcinus maenas on-site. We cordially thank Beate Johl (Zoological Institute and Museum at the University of Greifswald) for her assistance of analyzing the first antennae in specimens of the marine brachyurans of the Mediterranean Xantho hydrophilus, X. poressa, and Percnon gibbesi. We greatly appreciate the generous logistic and conceptual support by BS Hansson (Max Planck Institute for Chemical Ecology, Jena, Germany) and Mr. Mike Misso and Dr. Michael Smith from the Christmas Island National Park authorities during the collection of specimens of G. natalis. Glenda Jones and Sandra Banks are gratefully acknowledged for handling the research permits.

Additional Information and Declarations

Competing Interests

Author Contributions

Field Study Permissions

Data Availability

The authors declare there are no competing interests.

Jakob Krieger conceived and designed the experiments, performed the experiments, analyzed the data, wrote the paper, prepared figures and/or tables, reviewed drafts of the paper.

Philipp Braun performed the experiments, analyzed the data, wrote the paper, prepared figures and/or tables, reviewed drafts of the paper.

Nicole T. Rivera performed the experiments.

Christoph D. Schubart and Steffen Harzsch conceived and designed the experiments, contributed reagents/materials/analysis tools, wrote the paper, reviewed drafts of the paper.

Carsten H.G. Müller contributed reagents/materials/analysis tools, prepared figures and/or tables, reviewed drafts of the paper.

The following information was supplied relating to field study approvals (i.e., approving body and any reference numbers):

1. Australian Government; Department of the Environment; Parks Australia; Christmas Island National Parks.

2. AU˙COM 2010-090-1.

Raw data of brain section series is available from https://www.morphdbase.de under the “media” tab under a combination of the short title “Krieger” and an identifier according to the species and ID of the specimen.

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
