# Peer review of "Comparative analyses of olfactory systems in terrestrial crabs (Brachyura): evidence for aerial olfaction?"

_PeerJ, doi:10.7717/peerj.1433_

## Round 0.1 · original submission · Minor Revisions

· Academic Editor

Minor Revisions

This is a very, very nice piece of work on a fascinating topic. The presentation and figures are excellent. The two reviewers make useful suggestions to change the text. The observation of sexually dimorphic neuropilar features is interesting but should be couched, as there is a small sample size. If the typos are corrected I would be very happy for this to be accepted.

·

Basic reporting

The PDF does not include any indications of what raw data will be available. There are no other reporting issues.

Experimental design

This paper meets all standards for experimental design.

Validity of the findings

The data are fine and generally well-presented. As noted above, the raw data is not available for review in the PDF, but I have no major concerns that I think need to be addressed by inspecting the raw data.

Comments for the author

There are some minor suggestions for corrections of typos and minor wording changes. The only more substantive issue is that I would appreciate some indication of why Figure 17 (with sections from different species paired together) is organized as it is.

·

Basic reporting

The article is written in an appropriate format in standard English. I am attaching a pdf of the Word document with tracked changes to suggest minor copyedits for clarity, including to figure/table legends (my software did not permit annotation of the original pdf). Figures are nicely presented. Sufficient background information has been provided to contextualize the work.

I did not see any details in the article regarding plans for the deposition of raw data.

Experimental design

The research appears to be original and conducted to a high technical standard, with sufficiently detailed methods to be reproducible.

Validity of the findings

Although the authors examined four species with varying degrees of terrestriality, it was not clear to me whether any of the presumed adaptations correlated with the degree of terrestriality, which would be a very interesting finding. If it was only the case that terrestrial species were more similar to each other than to the aquatic species, that could be stated more clearly in the text.

One of the two major findings (sexual dimorphism of the olfactory glomeruli in U. tangeri) is based on a single female individual. Although the authors do use weaker language ("suggest") in recognition of this limitation in the main text (but not the abstract), an additional sample should be analyzed to strengthen this conclusion if possible.

Again, I did not see any details in the article regarding plans for the deposition of raw data.

---

## Round 0.2 · accepted · Accept

· Academic Editor

Accept

All the changes requested have been made. This is a very nice piece of work on a fascinating topic.